# The Analysis of Platelet-Derived circRNA Repertoire as Potential Diagnostic Biomarker for Non-Small Cell Lung Cancer

**DOI:** 10.3390/cancers13184644

**Published:** 2021-09-16

**Authors:** Silvia D’Ambrosi, Allerdien Visser, Mafalda Antunes-Ferreira, Ankie Poutsma, Stavros Giannoukakos, Nik Sol, Siamack Sabrkhany, Idris Bahce, Marijke J. E. Kuijpers, Mirjam G. A. Oude Egbrink, Arjan W. Griffioen, Myron G. Best, Danijela Koppers-Lalic, Cees Oudejans, Thomas Würdinger

**Affiliations:** 1Department of Neurosurgery, Cancer Center Amsterdam, Amsterdam UMC, Vrije Universiteit Amsterdam, 1081 HV Amsterdam, The Netherlands; s.dambrosi@amsterdamumc.nl (S.D.); m.antunesferreira@amsterdamumc.nl (M.A.-F.); m.g.best@amsterdamumc.nl (M.G.B.); d.lalic@amsterdamumc.nl (D.K.-L.); 2Department of Clinical Chemistry, Cancer Center Amsterdam, Amsterdam UMC, Vrije Universiteit Amsterdam, 1081 HV Amsterdam, The Netherlands; allerdien.visser@amsterdamumc.nl (A.V.); a.poutsma@amsterdamumc.nl (A.P.); cbm.oudejans@amsterdamumc.nl (C.O.); 3Department of Genetics, Faculty of Science, University of Granada, 18071 Granada, Spain; sgiannoukakos@ugr.es; 4Bioinformatics Laboratory, Biotechnology Institute, Centro de Investigación Biomédica, PTS, 18100 Granada, Spain; 5Brain Tumor Center Amsterdam, Cancer Center Amsterdam, Amsterdam UMC, Vrije Universiteit Amsterdam, 1081 HV Amsterdam, The Netherlands; ni.sol@amsterdamumc.nl; 6Department of Neurology, Cancer Center Amsterdam, Amsterdam UMC, Vrije Universiteit Amsterdam, 1081 HV Amsterdam, The Netherlands; 7Department of Physiology, Cardiovascular Research Institute Maastricht, Maastricht University Medical Center, 6229 ER Maastricht, The Netherlands; s.sabrkhany@maastrichtuniversity.nl (S.S.); m.oudeegbrink@maastrichtuniversity.nl (M.G.A.O.E.); 8Department of Pulmonary Diseases, Cancer Center Amsterdam, Amsterdam UMC, Vrije Universiteit Amsterdam, 1081 HV Amsterdam, The Netherlands; i.bahce@amsterdamumc.nl; 9Department of Biochemistry, Cardiovascular Research Institute Maastricht, Maastricht University, 6229 ER Maastricht, The Netherlands; marijke.kuijpers@maastrichtuniversity.nl; 10Angiogenesis Laboratory, Department of Medical Oncology, Amsterdam UMC, Vrije Universiteit Amsterdam, 1081 HV Amsterdam, The Netherlands; a.griffioen@amsterdamumc.nl

**Keywords:** circular RNA, non-small cell lung cancer, platelets, liquid biopsy, biomarkers

## Abstract

**Simple Summary:**

Interaction between blood platelets and cancer cells play an important role in various steps of cancer development and progression. These interactions lead to changes in the platelets’ RNA content, resulting in tumor-mediated “education” of platelets. Tumor-educated platelets (TEPs) can be used as a non-invasive biomarker source for cancer detection and progression monitoring. Our lab has previously identified that spliced mRNA TEP signatures provide specific information on the presence, location, and molecular features of cancers. Next to mRNA, other RNA types are present in platelets, and their repertoire can potentially be subjected to cancer-mediated alterations. Despite the evidence that circRNA could be a promising cancer biomarker, they have not yet been analyzed in blood platelets of cancer patients. In this proof-of-concept study, we aim to evaluate whether platelets’ circRNA signature could be used as a biomarker for cancer detection and progression.

**Abstract:**

Tumor-educated Platelets (TEPs) have emerged as rich biosources of cancer-related RNA profiles in liquid biopsies applicable for cancer detection. Although human blood platelets have been found to be enriched in circular RNA (circRNA), no studies have investigated the potential of circRNA as platelet-derived biomarkers for cancer. In this proof-of-concept study, we examine whether the circRNA signature of blood platelets can be used as a liquid biopsy biomarker for the detection of non-small cell lung cancer (NSCLC). We analyzed the total RNA, extracted from the platelet samples collected from NSCLC patients and asymptomatic individuals, using RNA sequencing (RNA-Seq). Identification and quantification of known and novel circRNAs were performed using the accurate CircRNA finder suite (ACFS), followed by the differential transcript expression analysis using a modified version of our thromboSeq software. Out of 4732 detected circRNAs, we identified 411 circRNAs that are significantly (*p*-value < 0.05) differentially expressed between asymptomatic individuals and NSCLC patients. Using the false discovery rate (FDR) of 0.05 as cutoff, we selected the nuclear receptor-interacting protein 1 (NRIP1) circRNA (circNRIP1) as a potential biomarker candidate for further validation by reverse transcription–quantitative PCR (RT-qPCR). This analysis was performed on an independent cohort of platelet samples. The RT-qPCR results confirmed the RNA-Seq data analysis, with significant downregulation of circNRIP1 in platelets derived from NSCLC patients. Our findings suggest that circRNAs found in blood platelets may hold diagnostic biomarkers potential for the detection of NSCLC using liquid biopsies.

## 1. Introduction

Circular RNAs (circRNAs) are a type of RNAs characterized by a covalently closed loop structure. They are generated by an alternative splicing mechanism called back splicing in which a downstream 5′ splice site binds an upstream 3′ splicing site of a pre-mRNA molecule [1]. Over the past few years, circRNAs have received increasing attention for their potential role as biomarkers for human disease and particularly in cancer [2]. Biomarkers can have different applications in the oncology field, such as diagnostic screening, the prognosis of the disease, treatment selection and response, and monitoring the progression of the disease [3]. CircRNAs have several features that make them a promising biomarker. A major characteristic of circRNAs is the absence of free 5′ and 3′ ends, which renders them resistant to exonucleases resulting in highly stable RNA molecules [4]. As a result, circRNAs have an increased half-life, compared to linear RNAs, even in blood circulation [5,6]. CircRNAs can be found in almost all human cell types analyzed thus far, and occasionally, they are more abundant than their linear counterparts [5,7]. Their high stability, abundance, and spatiotemporal specific expression [8] suggest that circRNAs may have a better analytical validity over canonical linear RNA as the biomarker.

Recent studies have shown that circRNAs are differentially expressed in different types of cancer, and they play a role in several steps of tumor initiation, proliferation, progression, and chemoresistance [9,10,11,12,13]. As an example, the expression of circZKSCAN1 [14], circSNAP47 [15], circDDX42 [16], and circARHGAP10 [17], found in tumor tissue biopsies or in clinically relevant cell lines, correlate with poor prognosis of lung cancer patients. Consequently, circRNAs may have clinical potential for cancer diagnosis, prognosis, and monitoring of treatment response.

CircRNAs are easily accessible through different body fluids, such as blood, urine, and saliva, and therefore suitable as biomarker candidates for liquid biopsy diagnostics [18].

Multiple studies have investigated the presence of circRNAs in the plasma of cancer patients [11,19,20,21,22,23,24]. In lung cancer, F-circEA is a circRNA generated by the back-splicing of the EML4-ALK fusion gene. While EML4-ALK mRNA can be detected only in the tumor tissues biopsies from non-small-cell lung cancer (NSCLC) patients, detection of F-circEA in plasma of those patients may serve as a potential indicator of EML4-ALK-positive NSCLC and may guide specific EML4-ALK-targeted therapy [20]. Another study performed with the plasma of 153 lung adenocarcinoma patients and 54 healthy controls has identified circYWHAZ and circBNC2 as potential biomarkers for lung cancer detection. The presence of both circRNAs led to an area under the curve (AUC) of 0.81 for the detection of lung cancer for all stages and an AUC of 0.83 for stage I patients [21]. circFARSA and circACP6 have also been identified as possible biomarker candidates for lung cancer detection in plasma with AUC of 0.71 and 0.79, respectively [22,23]. Moreover, a high level of hsa-circ-0000190 has been found in the plasma of lung cancer patients, correlated with high expression of PD-L1 in tumor tissue, and was shown to be associated with a poor response to systemic- and immunotherapy [24]. Cumulative results from these studies support the need to further explore the potential of circRNAs from blood samples as biomarkers for lung cancer diagnostics.

Apart from using plasma as a liquid biopsy source of potential cancer biomarkers, such as circRNAs, several other liquid biopsy biosources, including blood platelets and extracellular vesicles (including exosomes), are currently exploited as potent platforms for cancer diagnostics.

Blood platelets are small anucleate cells that originate from megakaryocytes and are the second most abundant cells in peripheral blood [25]. Platelets harbor several types of RNA (both coding and non-coding) that are inherited from megakaryocytes or directly and indirectly transfer from other cells [26,27,28]. The nucleic acid content of platelets can be affected or “educated” by their environment, including a primary tumor, by the passage of biomolecules or signaling that modified their RNA content [26,29]. The analysis of resulting alterations in the RNA profile of platelets can be employed as a biomarker for liquid biopsy diagnostics. Previously, we have shown that analysis of spliced RNA profiles from tumor-educated platelets (TEPs) enables the discrimination of patients with different types of cancer from healthy individuals or people with other pathologies [30,31,32,33].

Although circRNAs are found to be highly abundant in human platelets [34,35], currently, there is no analysis available addressing their biomarker potential in platelets samples collected from cancer patients. In the present study, we examined the circRNA content of platelets derived from patients diagnosed with NSCLC. This subtype of lung cancer represents approximately 85% of all lung cancer cases [36]. Despite the significant improvements in the oncological treatment of late-stage cancer, the survival rate still remains poor [37]. The discovery of sensitive and specific biomarkers is, therefore, a crucial step toward improved clinical management of patients with lung cancer.

In this proof-of-concept study, we investigated whether circRNAs derived from platelets may function as novel blood-based biomarkers for NSCLC detection.

## 2. Materials and Methods

### 2.1. Sample Collection and Study Population

Peripheral whole blood was drawn from NSCLC cancer patients and asymptomatic individuals at the Amsterdam UMC, VU University Medical Center, Amsterdam, The Netherlands, and Maastricht University Medical Center, Maastricht, The Netherlands.

Cancer patients were diagnosed by histological examination of tumor tissue and were confirmed to have tumor load at the moment of blood collection. Asymptomatic individuals had no medical history of diagnosis with any type of cancer prior to and/or at the moment of the blood collection. No additional tests were performed to confirm the absence of cancer.

Clinical data of the patients were collected such as patient age, gender, smoking habits, type of tumor, metastasis status, and current or previous treatments. Age- and gender matching were performed during the selection of the samples by including asymptomatic controls, and NSCLC samples with similar median age and gender distribution between the two groups. Blood from cancer patients was collected at the time of the diagnosis or one day before surgery in case of surgically removable (resectable) tumors.

At the Amsterdam UMC whole blood was collected in EDTA-coated purple-capped BD Vacutainer tubes containing the anticoagulant EDTA. Blood samples collected at the Maastricht University Medical Center were drawn in BD Vacutainer tubes containing 3.2% buffered sodium citrate. Both collection methods ensure to have minimal platelet-activating effects [30,38,39,40].

Clinical follow-up of asymptomatic controls is not available due to the anonymization of these samples according to the ethical rules of the hospitals.

This study was conducted in accordance with the principles of the Declaration of Helsinki. The medical ethics committee of both participating hospitals has approved this study. All the participants have received and signed the informed consent for blood collection and blood platelets analysis.

### 2.2. Sample Processing

Two protocols were used to isolate platelets. At the Amsterdam UMC, VU University Medical Center, the whole blood tube was centrifuged at 120× *g* for 20 min to separate platelet-rich plasma (PRP) from nucleated blood cells. PRP was then centrifuged at 360× *g* for 20 min to pellet the platelets. Both centrifugations were performed at room temperature. Platelet pellets were re-suspended in RNAlater (Thermo Scientific, Waltham, MA, USA), incubated overnight at 4 °C, and stored at −80 °C.

At the Maastricht University Medical Center, the blood tube was centrifuged at 240× *g* for 15 min to obtain PRP. Iloprost (50 nM) was added to PRP to minimize ex vivo platelet activation. PRP was centrifuged for two minutes at 1600× *g* to spin down the platelets. Both centrifugations were performed at room temperature. RNAlater was added to the platelets pellet and stored at −80 °C until use. Both protocols ensure the isolation of highly pure platelet pellets with minimum platelet activation and leukocyte contamination. No significant differences were observed between the two protocols.

### 2.3. RNA Isolation

Total RNA isolation was performed using mirVana miRNA isolation kit (Ambion, Thermo Scientific, cat nr. AM1560), according to the manufacturer’s instructions. The RNA was eluted in 30 μL of Elution Buffer. RNA quality and quantity were assessed using RNA 6000 Picochip (Bioanalyzer 2100, Agilent, Santa Clara, CA, USA). Only platelet RNA samples with a RIN-value greater than 7 and/or distinctive rRNA curves were included for analysis.

### 2.4. cDNA Library Construction

cDNA libraries were generated using 1 ng of total platelet RNA as input. SMARTer Stranded Total RNA-seq kit (Pico Input Mammalian User Manual, TakaraBio, Kyoto, Japan) was used for library preparation. In the protocol, the original reverse primer was substituted with N6-oligo primer lacking the oligo-dT primer. ZapR treatment with R-probes was performed to removed ribosomal and mitochondrial cDNA.

Final PCR amplification was performed with 16 cycles. Purifications were performed with Agencourt AMPure XP beads (A63881, Beckman Coulter, Brea, CA, USA) after the addition of Illumina adapters and after final RNA-seq library amplification.

### 2.5. RNA Sequencing

Paired-end sequencing (2 × 150 bp) of the samples was performed on the Illumina HiSeq 4000 using version 4 of the TruSeq reagents. Index combinations were chosen according to the Illumina recommendations. An equimolar pool of 12 samples was sequencing per line of the flowcell. As a control, PhiX control library (Illumina, San Diego, CA, USA) was spike-in at 5%.

### 2.6. RNA Sequencing and Data Processing

Files were demultiplexed using bcl2fastq software (v2.20, Illumina, San Diego, CA, USA). Adapter sequences were removed using Cutadapt-v2.3. The quality of the fastq files was assessed by using the FastQC-v0.11.8 software. HISAT-v2.1.0 was used to map the sequenced reads against an in-house generated from the University of California Santa Cruz (UCSC) reference, which is based on the human reference genome (hg38). After sorting for name and chromosome, followed by indexing with Samtools-v1.9, consistency, and quality of .bam files were checked using Integrative Genomics Viewer-v2.5.3.

### 2.7. CircRNA Analysis and Identification of DE circRNA

CircRNA identification and analysis were performed as described by Oudejans et al. [41]. *Awk* command was used to recover unmapped reads from the Hisat-v2.1.0 output file with flags 73, 69, and 77 in read 1, and 133, 137, and 141 in read 2. The recovered reads were then analyzed with Accurate circRNA Finder Suite (ACFS) in cohort mode, in order to identify and quantify the circRNA present [42]. Settings were Seq_len, 150; BWA_seed_length, 19; BWA_min_score, 30; Thread (number of processors—1); minJump, 100; maxJump, 1,000,000; minSplicingScore, 10; minSampleCnt (n); minReadCnt, 2; minMapping Quality, 30; Coverage, 0.9; minSpanJunc, 35; ErrorRate, 0.01; and Strandness -. Settings (n) for minSampleCnt corresponded to 25% of the total number of samples analyzed. For ACFS analysis in cohort mode, a high-performance cluster was used. UCSC hg38 was used for the annotation of the ACFS.

The read count table generated was used as input for the thromboSeq.R script (https://github.com/MyronBest/thromboSeq_source_code, accessed on 19 November 2018) using RStudio-v1.2.5001 [38]. The following settings were used: FDR: false/true, with or without PSO enhancement, variable minimal number of counts in 90% of samples, variable cross-correlation settings.

The script outputs two heatmaps, with and without particle swarm optimization (PSO) enhancement, age and gender box plots, and the ANOVA table with included log fold change (logFC), log counts per million (logCPM), likelihood ratio (LR), *p*-value, and false discovery rate (FDR) for each circRNA.

### 2.8. Reverse Transcription–Quantitative PCR Analysis of circRNAs

Primer sequences for circNRIP1 and circMAN1A2 are described in Appendix A. Primers were designed to amplify the back splicing junction of the circRNA of interest. Primers and probes were designed accordingly to the “Primer and TAMRA dye-labeled Probe Design” quantification guidelines, with the only exception that the probe length parameter was set at 20–30 bases. The different primers pair combinations were checked for specificity against the human genome and for the absence of primer-dimers and repeats. Probes were labeled with 5′TAMRA and 3′DQ.

XpressRef Universal RNA (QIAGEN, Hilden, Germany) was used as reference RNA, which contains total RNA from 20 different human adult and fetal tissues. Different input concentrations of reference RNA (1, 10, and 100 ng/uL), primers, and probes were used to optimized the reaction.

The same reference RNA was also used as a positive control of the experiment and as a negative control, we substituted the RNA with Milli-Q water.

Specific cDNA construction was performed using Maxima H Minus First Strand cDNA Synthesis Kit (K1652, Thermo Fisher Scientific, Waltham, MA, USA). Platelets/reference RNA (1 ng), specific reverse primer (circNRIP1_primer_Rv or circMAN1A2_primer_Rv, 20 pmol), deoxynucleotide (dNTP, 1 mM) and Milli-Q water (total volume of 15 μL) were mixed together and incubated 5 min at 65 °C. Then, 5X RT Buffer and Maxima H Minus Enzyme Mix were added to the previous mix (final volume 20 μL) and incubated for 30 min at 50 °C and 5 min at 85 °C. TaqMan PreAmp Master Mix was used to pre-amplify the cDNA generated. TaqMan PreAmp Master Mix (25 μL) was mixed with 10 μL of cDNA, 12.5 μL of pooled assay mix (mix of forward, reverse primers, and probes, with the final concentration of each 0.5 pmol/μL) and Milli-Q water (final volume 50 μL). The reaction was incubated 10 min at 95 °C for the activation of the enzyme, followed by 15 s at 95 °C, 4 min at 60 °C for 14 cycles, and 10 min at 99 °C to inactive enzymatic reaction.

The automated QIAgility system and Rotor-Gene-Q platform were used for automated PCR setup and quantitative real-time PCR cycling, respectively. Each reaction was performed in triplicate. Primers and probe concentrations are described in Appendix A. The real-time qPCR reaction was performed for two minutes at 50 °C, 20 s at 95 °C, and then 40 cycles of 30 s at 95 °C and 30 s at 60 °C. For the analysis of the quantitative expression of circRNA NRIP1, we used circRNA MAN1A2 as a housekeeping gene. We normalized each run using the reference RNA (XpressRef Universal RNA, QIAGEN, Germany) and applied statistical analysis by unpaired non-parametric *t*-tests with Welch’s correction using a two-tailed *p*-value (GraphPad Prism-v.8.0.2, GraphPad, San Diego, CA, USA). Equal standard deviations were not assumed since an independent cohort of samples was used.

## 3. Results

### 3.1. Profiling of Platelets circRNA Repertoire by RNA Sequencing and Differential Expression Analysis

To investigate whether the circRNA profile in platelets may hold the potential as non-invasive biomarkers for NSCLC detection, we performed RNA-sequencing of 12 platelet samples derived from both NSCLC patients (*n* = 6) and asymptomatic individuals (*n* = 6) (Table 1, Figure 1a, Appendix A). Platelets collected in both hospitals were equally distributed between the two groups and samples were age- and gender matched. Both metastasized and localized NSCLC platelets samples were included in this exploratory experiment (Table 1). Total RNA from platelets was extracted and checked for quality (Appendix A), before library preparations and RNA sequencing (Appendix A).

After processing of raw sequencing data, differential expression analysis of circRNAs was performed, using the thromboSeq software with adjustment of different settings, i.e., FDR: false/true, with or without PSO enhancement, variable minimum number of counts in 90% of samples, variable cross-correlation settings [38]. To evaluate the performance of the analysis we investigated (1) the presence of Plt-circR4 (chr4:104,519,454-104,518,577), a platelet-specific biomarker, as positive control [41]; (2) segregation of the two groups (tumor vs. control) using heatmap-dendrogram clustering; (3) identification of potential tumor-specific biomarker using FDR < 0.05 as cutoff; (4) validation of circRNAs with FDR < 0.05 in an independent cohort of samples by reverse transcription–quantitative PCR (RT-qPCR).ACFS could correctly predict the presence of Plt-circR4, a novel platelet-specific circRNA (criterion #1) confirming the validity of the mapping (annotation) approach applied. Based on this analysis, we were able to identify 4732 circRNAs in all sequenced samples. ACFS-thromboSeq pipeline using two-way ANOVA allows the generation of the heatmap with PSO enhancement, which shows samples distribution (clustering) according to their original group assigned (criterion #2) (NSCLC vs. asymptomatic individuals (termed as control), Figure 1b). The corresponding ANOVA output with differentially distributed circRNAs (Appendix A) resulted in the identification of 411 differentially expressed (*p* < 0.05) circRNAs, of which 327 were downregulated and 84 upregulated (Figure 1c) in NSCLC. Further analysis based on the FDR value below 0.05 as cutoff resulted in the selection of one circRNA that belongs to the *NRIP1* gene located at the human chromosome 21 (chr21:15,043,574-15,014,344) (Figure 2a). Between the 10 circRNA variants derived from the *NRIP1* gene predicted, only one showed highly significant downregulation in the NSCLC samples (*p* = 6.49 × 10^−10^; FDR = 3.07 × 10^−6^) (criterion #3). This circRNA derived from exon 2 and 3 of the 5′untraslated region of the *NRIP1* gene.

### 3.2. Validation of NRIP1 circRNA by RT-PCR

To investigate whether NRIP1 circRNA (circNRIP1; chr21:15,043,574-15,014,344, Figure 2a) could be employed as a potential biomarker for NSCLC, we analyzed the expression by RT-qPCR (criterion #4) in an independent cohort of platelet samples collected from 23 NSCLC patients and 24 asymptomatic individuals (Table 2, Appendix A). The circRNA expression levels were normalized using circRNA MAN1A2 expression as the housekeeping gene [41]. The relative expression of circNRIP1 is calculated using the comparative 2^−ΔΔCt^ method [43,44]. RT-qPCR approach detected a significant downregulation of circNRIP1 in the NSCLC group, compared with our control group (*p*-value = 0.0302) (Figure 2b). This observation is consistent with the results obtained from the analysis of the RNA-sequencing data.

Further statistical analysis showed that the downregulation of circNRIP1 is correlated with the advanced (late-) stage of the disease. Dividing the NSCLC group in samples derived from patients with resectable tumor (early stage NSCLC; *n* = 11) and metastasized tumor (late-stage NSCLC; *n* = 12), we observed a significant downregulation of circNRIP1 only in the late-stage NSCLC group (*p*-value = 0.0236), as compared to the asymptomatic individuals. The expression level of circNRIP1 had no statistically significant (*p*-value = 0.098) change between early stage NSCLC (*n* = 11) and asymptomatic individuals (*n* = 24) and between early stage NSCLC and late-stage NSCLC (*n* = 12, *p*-value = 0.417) (Figure 2c). We also evaluated whether other confounding factors, such as smoking habits and histological subtypes of NSCLC, could influence the expression level of circNRIP1. There were no significant differences between the group analyses (Appendix A), confirming that the significant downregulation of circNRIP1 is associated with late-stage NSCLC.

## 4. Discussion

Multiple studies have shown that the RNA repertoire of circulating blood platelets is highly affected by pathological conditions such as cancer, providing the opportunity to use platelet-derived RNAs as diagnostic, prognostic, and predictive biomarkers [30,31]. Next to the already established RNA-based biomarkers such as mRNAs and microRNAs, the circRNAs are emerging as a novel biomarker candidate. Platelets have a higher circRNA content, compared with their progenitor cells, megakaryocytes, or other hematopoietic cells [27,34,35]. The enrichment of circRNAs may be due to the degradation of linear RNAs occurring in platelets [35]. Although circRNA could represent a promising biomarker for liquid biopsy, the potential to detect NSCLC based on their profile in platelets was not investigated yet.

We examined the circRNA profile of platelets of NSCLC patients and asymptomatic individuals using high-throughput RNA-seq. We identified a total of 4732 circRNAs of which 84 circRNAs were found significantly upregulated (*p*-value < 0.05) and 327 significant downregulated (*p*-value < 0.05) in platelets derived from NSCLC patients. These findings indicate that the circRNA transcriptome of platelets may be altered in the presence of cancer.

Dysregulation of circRNA in platelets derived from NSCLC patients complements several studies addressing the alteration of platelet RNA content in response to cancer cell signals including the uptake of cancer-related mutant RNA and fusion transcripts [45,46].

Out of the 411 differentially expressed circRNAs identified, only 1 presented an FDR below 0.05. This circNRIP1 sequence, covering the exon 2 and 3 of the 5′-untranslated region of the transcript (Figure 2a) was significantly downregulated in platelets derived from NSCLC patients, compared with those from asymptomatic individuals. NRIP1 is a receptor-interacting protein that functions as a coregulator by activating or repressing different receptor transcription factors. Its function is not completely understood, but it is known that NRIP1 can corepress transcription factors as thyroid hormone receptor (TR), estrogen-related receptor (EER), and E2F transcription factor 1 (E2F1) [47]. Dysregulation of this gene has been observed in different human pathological conditions, such as in cancer patients and in preeclamptic pregnancies [48,49]. circNRIP1 has been found differentially expressed in several types of tumor tissues, such as gastric cancer, cervical cancer, and ovarian cancer [50,51,52,53]. circNRIP1 regulates pathways involved in cell proliferation, migration, and invasion [50,52,53,54] and may also be involved in chemoresistance mechanisms, representing a possible target for cancer resistance treatments [51,55].

Its role in circulating platelets in the presence of cancer is currently unknown. The validation of circNRIP1 expression in an independent cohort of platelets samples revealed a possible correlation between the downregulation of circNRIP1 and the advanced (late) stage of the disease, suggesting circNRIP1 as a potential biomarker of lung cancer progression.

Similar observations were also found in tissue and plasma of gastric cancer patients, in which downregulation of circNRIP1 was significantly associated with the advanced stage of the disease [52]. A second study has also shown that gastric cancer cells can transmit circNRIP1 via exosome to other cancer cells to promote tumor metastasis in vivo [54].

Interestingly, circNRIP1 has been also observed to be highly abundant in platelet-derived exosomes [35]. This may suggest a possible release of circNRIP1 from platelets through the exosome to promote cancer progression and metastasis. Very little is known about platelet-derived extracellular vesicles (EVs), including exosomes, and their content, especially in cancer. In addition, platelet-released EVs may partly remain associated with the outer cell membrane [56]; thus, their RNA content could potentially be co-isolated during platelet RNA extraction. Our current method of platelets isolation has only been optimized for platelets purity from erythrocytes and leukocytes [38]; therefore, we cannot exclude any cross contamination with RNA material derived from membrane-bound EVs. Future studies should address if platelet-derived EVs might function as a communication vehicle between cells including cell fragments such as platelets by transmitting circRNAs to promote cancer progression.

Cumulative data from this study has shown that platelet-derived circRNAs may hold the potential to be further exploited as cancer biomarkers for liquid biopsy diagnostics. However, the drawback we currently encountered is the limited number of platelets samples sequenced for this proof-of-concept study. Increasing the number of samples will provide more robust statistical power, required for the selection and clinical validation of circRNA biomarkers. The RT-qPCR experiment confirmed the downregulation of circNRIP1 in late-stage NSCLC platelets samples, suggesting its potential role as an indicator for cancer progression. Although the role of circNRIP1 has been studied in different types of cancers, future work may underpin its function in the advanced stage of NSCLC. In addition, circNRIP1 as a biomarker for cancer progression and metastatic formation needs to be further investigated in platelets and other types of liquid biopsy biosources, such as EVs, including exosomes. Moreover, integration of different RNA species in the analysis, such as combining data on circRNA and mRNA from different biosources, in a multimodal analysis approach, could further advance the diagnostic potential of liquid biopsies for NSCLC.

## 5. Conclusions

In conclusion, in this proof-of-concept study, we provided lines of evidence that platelet-derived circRNA repertoire changes in the presence of NSCLC. The differential circRNA expression signature found between cancer patients and asymptomatic individuals may be used to diagnose and monitoring NSCLC status by minimally invasive blood analysis.

Moreover, we identified circNRIP1 as a possible biomarker candidate for lung cancer progression.

## Figures and Tables

**Figure 1 cancers-13-04644-f001:**
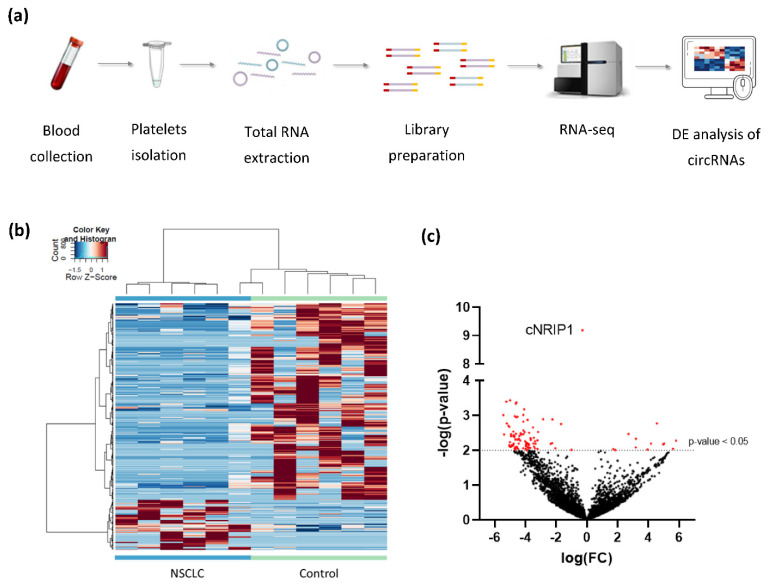
Differential expression analysis of platelet-derived circRNAs distinguishes patients with NSCLC from asymptomatic individuals: (**a**) schematic representation of the workflow. Whole blood samples were subject to platelets isolation and total RNA extraction. Libraries were generated using SMARTer Stranded Total RNA-Seq Kit using N6-oligo primer lacking the oligo-dT primer. cDNA is sequenced using high-throughput paired-end sequencing on the Illumina HiSeq 4000 platform; (**b**) hierarchical clustering of differentially expressed circRNAs between NSCLC (*n* = 6; green) and asymptomatic individuals (control; *n* = 6; blue). Clustering was performed with PSO enhancement. Samples are indicated in the columns, circRNAs are indicated in the rows, and color intensity represents the Z score-transformed expression values; (**c**) volcano plot of differentially expressed circRNAs. The negative log of the *p*-value (base 10) is plotted on the *Y*-axis, and the log of the FC (base 2) is plotted on the *X*-axis. Red dots indicate significantly differentially expressed circRNAs (*p*-value < 0.05).

**Figure 2 cancers-13-04644-f002:**
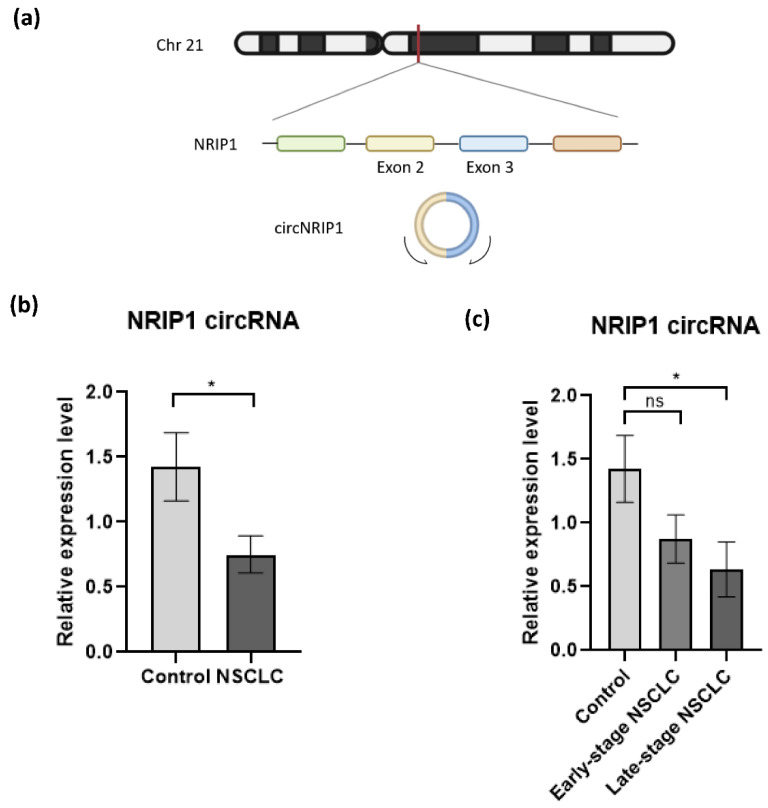
Validation of circNRIP1 expression in platelet-derived RNA. (**a**) Schematic illustration showing that circNRIP1 is formed by head-to-tail splicing of NRIP1 exon 2 and exon 3; (**b**) the expression of two circRNAs, NRIP1 and MAN1A2 in platelet samples derived from NSCLC patients (*n* = 23) and asymptomatic individuals (*n* = 24) was analyzed by RT-qPCR. The MAN1A2 circRNA was used as endogenous circRNA control for normalization as the “housekeeping gene”. Each sample was analyzed in triplicate. All data are presented as mean ± SEM. The 2^−ΔΔCt^ method was used to calculate the relative expression of circNRIP1 between the different groups. For the statistical analysis, we used the unpaired nonparametric *t*-test with Welch correction with two-tailed *p*-value; (**c**) The expression level of circNRIP1 is significantly downregulated (*p*-value = 0.0302) in platelets of NSCLC patients, specifically in samples from patients diagnosed with late-stage disease (*p*-value = 0.0263, *n* = 12). Statistically non-significant (indicated as ns) downregulation is observed in patients with early stage NSCLC (*p*-value = 0.098, *n* = 11). * indicates *p*-value < 0.05.

**Table 1 cancers-13-04644-t001:** Data table with the general characteristics of sample cohort (asymptomatic individuals and NSCLC patients) including sampling location of blood platelets employed for the circRNA analysis by RNA-Seq.

N° of Samples	Control	NSCLC
	6	6
Median age (Min–Max) (years)	68 (63–80)	66 (56–80)
F/M (%)	67/33	67/33
Localized (%)	-	33
Metastasized (%)	-	67
Hospital #1	3	3
Hospital #2	3	3

F, female; M, male; Hospital #1, Maastricht University Medical Center; Hospital #2, Amsterdam UMC, VU University Medical Center.

**Table 2 cancers-13-04644-t002:** Data table with the general characteristics of sample cohort (asymptomatic individuals and NSCLC patients) including sampling location of blood platelets employed for the circRNA analysis by RT-qPCR. F, female; M, male; Hospital #1, Maastricht University Medical Center; Hospital #2, Amsterdam UMC, VU University Medical Center.

N° of Samples	Control	NSCLC
	24	23
Median age (Min–Max) (years)	62.9(51–79)	60.2 (41–78)
F/M (%)	42/58	39/61
Localized (%)	-	48
Metastasized (%)	-	52
Hospital #1	24	11
Hospital #2	0	12

## Data Availability

Raw files will be deposited at Gene Expression Omnibus once the manuscript has been accepted for publication. During the review process, the raw file data can be provided upon request.

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
