# Peer review of "The Analysis of Platelet-Derived circRNA Repertoire as Potential Diagnostic Biomarker for Non-Small Cell Lung Cancer"

_cancers, 2021, doi:10.3390/cancers13184644_

Round 1

Reviewer 1 Report

The manuscript could benefit from a clearer exposition and a more focused discussion; moreover, the discussion somehow seems to fail to interpret the data in the context of what is known in the field: it actually sounds redundant, as it largely summarizes again data already presented in the Results without properly placing them in the proper scientific context.

The following pertinent articles should be mentioned:

PMID: 33668203 

PMID: 32460515

PMID: 30087655 

PMID: 33807355 

PMID: 34290668 

PMID: 33076269

PMID: 32576868

PMID: 33466455 

PMID: 29657280 

The strengths and limitations of the study should be deeply addressed, taking into account sources of potential bias or imprecision: Discuss both direction and magnitude of any potential bias.

Author Response

Response to Reviewer 1 Comments

The manuscript could benefit from a clearer exposition and a more focused discussion; moreover, the discussion somehow seems to fail to interpret the data in the context of what is known in the field: it actually sounds redundant, as it largely summarizes again data already presented in the Results without properly placing them in the proper scientific context.

The following pertinent articles should be mentioned:

PMID: 33668203 

PMID: 32460515

PMID: 30087655 

PMID: 33807355 

PMID: 34290668 

PMID: 33076269

PMID: 32576868

PMID: 33466455 

PMID: 29657280 

The strengths and limitations of the study should be deeply addressed, taking into account sources of potential bias or imprecision: Discuss both direction and magnitude of any potential bias.

Response:

We thank the reviewer for her/his constructive comments.

We now integrated in the manuscript the papers that the reviewer suggested (references 2, 11,12,13,27 and 28).

We revised the discussion section to clarify our findings and added further important works in support of our thesis. For instance, we now included additional studies regarding the role of circNRIP1 in different types of cancer and its role as potential biomarker for advanced disease in gastric cancer.

Although this manuscript is a proof-of-concept study, the main drawback of this work is the limited number of samples used, which decreases the statistical power of the analysis for the selection and validation of the circRNA biomarkers (discussed in lines 492-494 of the Discussion section). Furthermore, possible cross-contamination of RNA material derived from platelet-derived EVs during the process of platelets isolation is also addressed in the discussion section (lines 480-489). Confounding variables influences, such as blood collection locations, smoking habits, histological subtypes, age and gender, were excluded from our analysis as described in the Method and Result sections (lines 138-142, 278-279 and 383-387).

Overall, this manuscript describes a method for identification and quantification of circRNAs from platelet RNA sequencing data. The validation by RT-qPCR methodology has confirmed the expression level of the only circRNA biomarker identified with FDR below 0.05. Increasing the number of samples may provide us with more statistical power for identification of additional circRNA biomarkers (Discussion section, lines 493-494).

Reviewer 2 Report

The work by D’Ambrosi et al is based on previous studies from the same group which identified that spliced mRNA TEP signatures provide specific information on the presence, location and molecular features of cancers. Next to mRNA, other RNA types are present in platelets and their repertoire can potentially be subjected to cancer-mediated alterations. Based on the evidence that circRNA could be a promising cancer biomarker, the authors analyzed circRNAs in blood platelets of cancer patients and provide proof-of -concept on whether platelets circRNA signature could be used as a biomarker for cancer detection and progression. The study is well designed, and sound and the results suggest that at least on circRNA (namely circNRIP1) was verified as downregulated in the analysis as well as in a cohort of patients and normal individuals. The manuscript is well written and presented, however I feel that some points require further clarifications. More specifically:

1.       It would useful for the readership if the authors could provide a brief table of the most significant alterations after FDR analysis (i.e. the 5 or 10 most upregulated and downregulated circRNAs). Although the provide a table in the supplementary material, the majority of the reported circRNAs appear downregulated. The authors should clarify this point.

2.       The authors should clarify in the text why they chose circNRIP1 as the sole circRNA to verify. It would be also useful to provide any additional analysis that they might have on upregulated circRNAs. For example, DLG1 could be a good candidate from what they present with established role in cell migration and proliferation.

3.       The authors should discuss at least some part of their data in comparison to circRNAs that have been identified in biopsies, or after treatment. The role of circNRIP1 should be discussed from this point of view.

4.       The authors mention that detected 10 different variants of circNRIP1. The authors should describe the differences for the general readership and discuss briefly why they are produced (or their possible importance). They should also clarify the qRT-PCR verification into more detail, since to my understanding their design of primers could not discriminate different variants.    

Author Response

Response to Reviewer 2 Comments

The work by D’Ambrosi et al is based on previous studies from the same group which identified that spliced mRNA TEP signatures provide specific information on the presence, location and molecular features of cancers. Next to mRNA, other RNA types are present in platelets and their repertoire can potentially be subjected to cancer-mediated alterations. Based on the evidence that circRNA could be a promising cancer biomarker, the authors analyzed circRNAs in blood platelets of cancer patients and provide proof-of -concept on whether platelets circRNA signature could be used as a biomarker for cancer detection and progression. The study is well designed, and sound and the results suggest that at least on circRNA (namely circNRIP1) was verified as downregulated in the analysis as well as in a cohort of patients and normal individuals. The manuscript is well written and presented, however I feel that some points require further clarifications. More specifically:

Point 1.       It would useful for the readership if the authors could provide a brief table of the most significant alterations after FDR analysis (i.e. the 5 or 10 most upregulated and downregulated circRNAs). Although the provide a table in the supplementary material, the majority of the reported circRNAs appear downregulated. The authors should clarify this point.

Response:

We thank the reviewer for the comment. In the supplementary Table S1, we provided the list of the 30 most differential expressed circRNAs ordered by increasing p-value. In our analysis we found a total of 411 significantly differentially expressed circRNAs (p-value <0.05), of which 327 were downregulated. This trend is also reflected in supplementary table S1 where 28 out of the 30 most differently expressed circRNAs are downregulated.

We now specify in the supplementary Table S1 legend the total of most 30 significant differentially expressed circRNAs and their order:

“Supplementary Table S1.  ANOVA output of the top 30 circRNAs (out of 411) differentially expressed in platelets of NSCLC patients (n=6) versus platelets of asymptomatic individuals (n=6), ordered by increasing p-value. The origin of the circRNAs is provided by chromosome number followed by position of back-splice junctions (GRCh38/hg38), positive (+) or negative (-) strand and size of genomic region. LogFC: logarithm of fold change, logCPM: logarithm of counts per million, LR: likelihood ratio, p-value: probability value, FDR: p-value corrected for false-detection rate”

Point 2.       The authors should clarify in the text why they chose circNRIP1 as the sole circRNA to verify. It would be also useful to provide any additional analysis that they might have on upregulated circRNAs. For example, DLG1 could be a good candidate from what they present with established role in cell migration and proliferation.

Response:

The criterium used for the selection of potential circRNA biomarker was an FDR value <0.05. circNRIP1 was the only circRNA with FDR <0.05. We indicated the selection criterium in the text, in line 289-291: “(3) identification of potential tumor-specific biomarker using FDR <0.05 as cut-off; (4) validation of circRNAs with FDR <0.05 in an independent cohort of samples by reverse transcription-quantitative PCR (RT-qPCR).” and in lines 310-312: “Further analysis based on an FDR threshold <0.05 resulted in the selection of a single circRNA that belongs to the NRIP1 gene located at human chromosome 21 (chr21:15043574-15014344)”.

Although, DLG1 was significantly upregulated (p-value <0.05), the FDR value did not pass the threshold and therefore we did not include it in further validation by RT-qPCR.

As mentioned in the manuscript, increasing the number of samples may provide more robust statistical power for the selection of additional circRNA biomarker candidates. We cannot exclude that by increasing the cohort of samples DGL1 could be included as an additional possible circRNA biomarker candidate.

Point 3.       The authors should discuss at least some part of their data in comparison to circRNAs that have been identified in biopsies, or after treatment. The role of circNRIP1 should be discussed from this point of view.

Response:

We agree with the reviewer and we now added additional information on further studies of circNRIP1 performed in tissue biopsies (line 455-459): “circNRIP1 has been found differentially expressed in several types of tumor tissues, such as gastric cancer, cervical cancer and ovarian cancer [51]–[54]. circNRIP1 regulates pathways involved in cell proliferation, migration and invasion [51], [53]–[55] and may also be involved in chemoresistance mechanisms, representing a possible target for cancer resistance treatments [52], [56].”

In addition, we add in line 467-469:” Similar observations were also found in tissue and plasma of gastric cancer patients, where downregulation of circNRIP1 was significantly associated with the advanced stage of the disease [53].”

Point 4.       The authors mention that detected 10 different variants of circNRIP1. The authors should describe the differences for the general readership and discuss briefly why they are produced (or their possible importance). They should also clarify the qRT-PCR verification into more detail, since to my understanding their design of primers could not discriminate different variants. 

Response:

We thank the reviewer for the interesting comment. At this time, we can only speculate about the function of the different splice variants of circRNAs in general, and that of NRIP1 in particular. The circNRIP1 variant with differential expression in NSCLC is specific for the 5’-untranslated region. A function related to translational control such as competition or interaction with RNA binding proteins and/or other factors with a translational quantitative effect on RNA such as miRNA are likely.

Regarding the specificity of the primers used for the qRT-PCR, they have been previously tested and described (Clin Chem, Oudejans et al.). Primers were designed to confirm the presence of the back-splicing junction of circNRIP1 (3’end of exon 3 – 5’end of exon 2). Specificity of the primers was confirmed by 2 negative control reactions: (1) with no reverse transcription enzyme and (2) input material replaces with water. RT-qPCR fragment were analyzed by the QIAxcel capillary electrophoresis system and by Sanger sequencing of the purified amplicon (Clin Chem, Oudejans et al.).

Reviewer 3 Report

In this Case-Control study, the authors explored for potential liquid biomarkers focusing on the platelet-derived circRNA to predict the presence of NSCLC. The authors applied RNA seq for 6 pairs of NSCLC patients and healthy controls (a discovery cohort) and identified that downregulated cirNRIP1 expression was a potential biomarker for NSCLCs. The authors then examined an independent validation cohort (n = 47) to test the expression levels of cirNRIP1 by RT-PCR. The reviewer thinks the study was well performed, but raises a few major comments as summarized below.

  1. Histological data (adenocarcinoma, squamous cell carcinoma, or others) should be added. Can cirNRIP1 downregulation be a potential biomarker for both types of NSCLCs?
  2.  The reviewer thinks smoking data is also important for both of the NSCLC cohort and the healthy control, since smoking may affect the expression of genes of platelets.
  3.  How about the cut-off value of cirNRIP1 to predict the presence of NSCLC? How about the AUC?

Reviewer 4 Report

The paper is very interesting.

Just a few comments concerning sample collection.

The 2 centres used different methods to centrifuge blood samples. This different operation could affect the amount of platelet recovered and the analysis performed.

Please add a comment in the discussion and methods about this issue.

Author Response

Response to Reviewer 4 Comments

The paper is very interesting.

Just a few comments concerning sample collection.

The 2 centres used different methods to centrifuge blood samples. This different operation could affect the amount of platelet recovered and the analysis performed.

Please add a comment in the discussion and methods about this issue.

Response:

We thank the reviewer for her/his comment.

Both protocols ensure the isolation of platelets at high level of purity and without inducing platelet activation, as described in Best et al. 2019, Sabrkhany et al. 2013 and Sabrkhany et al 2017. Multiple quality control steps were performed during platelet isolation, RNA isolation and sequence library preparation. No significant differences were observed between the two protocols. We now added to the text in the revised Method section the following sentence (lines 168-170): “Both protocols ensure the isolation of highly pure platelet pellets with minimum platelet activation and leukocyte contamination. No significant differences were observed between the two protocols.”. Moreover, a balanced number of samples derived from the two different institutions was used in our discovery cohort (see Table 1), and an equal amount of RNA input was used to prepare the sequencing libraries.